# Galangin and 1′-Acetoxychavicol Acetate from Galangal (*Alpinia galanga*) Suppress Lymphoma Growth via c-Myc Downregulation and Apoptosis Induction

**DOI:** 10.3390/biology14081098

**Published:** 2025-08-21

**Authors:** Sirinya Moakmamern, Lapamas Rueankham, Natsima Viriyaadhammaa, Kittikawin Panyakham, Natnicha Khakhai, Pinyaphat Khamphikham, Suwit Duangmano, Siriporn Okonogi, Sawitree Chiampanichayakul, Songyot Anuchapreeda

**Affiliations:** 1Department of Medical Technology, Faculty of Associated Medical Sciences, Chiang Mai University, Chiang Mai 50200, Thailand; sirinya_moakm@cmu.ac.th (S.M.); lapamas.rk96@gmail.com (L.R.); natsima.v@cmu.ac.th (N.V.); putermaneethon@gmail.com (K.P.); natnicha.khakhai@gmail.com (N.K.); pinyaphat.kha@cmu.ac.th (P.K.); suwit.du@cmu.ac.th (S.D.); 2Cancer Research Unit of Associated Medical Sciences (AMS CRU), Faculty of Associated Medical Sciences, Chiang Mai University, Chiang Mai 50200, Thailand; 3Department of Pharmaceutical Sciences, Faculty of Pharmacy, Chiang Mai University, Chiang Mai 50200, Thailand; okng2000@gmail.com; 4Center of Excellence in Pharmaceutical Nanotechnology, Chiang Mai University, Chiang Mai 50200, Thailand

**Keywords:** galangal, *Alpinia galanga*, galangin, 1′-acetoxychavicol acetate, zingiberaceae, cancer, Raji, Daudi, lymphoma, c-Myc, apoptosis

## Abstract

Lymphoma is a type of cancer that affects white blood cell function and often involves abnormal activity of a protein called c-Myc, which controls how cells grow and divide. This study explored whether certain traditional medicinal plants from the ginger family, including galangal (*Alpinia galanga*), black turmeric (*Curcuma aeroginosa*), black ginger (*Kaempferia parviflora*), phlai lueang (*Zingiber montanum*), and phlai dum (*Zingiber ottensii*), could help fight lymphoma. Extracts from these five plants were tested on two types of lymphoma cell lines (Raji and Daudi) and compared with their effects on normal blood cells. Among them, galangal showed the strongest ability to kill cancer cells while being less harmful to healthy cells. Further testing revealed that two natural compounds from galangal, especially one called 1′-acetoxychavicol acetate (ACA), worked by blocking the cancer-related c-Myc protein, stopping the cancer cells from growing, and triggering them to die. This research suggests that compounds from galangal, a commonly used spice and traditional remedy, may offer a promising and safer approach for developing new treatments for lymphoma. It provides a foundation for turning plant-based ingredients into cancer therapies that are more natural and potentially less toxic than current options.

## 1. Introduction

Lymphoma is the third most common cancer in children, following leukemia and tumors of the brain and other parts of the nervous system. It is a hematolymphoid malignancy that is frequently diagnosed worldwide, accounting for over 3% of all cancer cases. In 2019, lymphoma contributed to 4.2% of new cancer diagnoses and 3.3% of cancer-related deaths globally. Lymphoma is a cancer of the lymphatic system and is classified as a group of lymphoproliferative disorders originating from B-lymphocytes, T-lymphocytes, or natural killer (NK) cells. All lymphoid malignancies arise from systemic lymphoid tissues and have the potential to disseminate throughout the body. Hodgkin’s lymphoma accounts for approximately 10% of all lymphoma cases, while the remaining 90% are classified as non-Hodgkin lymphoma (NHL), which presents with diverse histological and clinical characteristics. In Thailand, between 2016 and 2018, NHL was the fifth most common cancer among males (6.6%) and in the ninth most common among females (5.2%) [1]. According to previous reports, B cell lymphomas are responsible for 85–95% of lymphomas [2]. The most prevalent type of NHL in Thailand is diffuse large B-cell lymphoma (DLBCL, NOS), accounting for 58.1% of cases [3].

B lymphocytes are a type of white blood cells involved in the immune system, responsible for producing antibodies and originating from stem cells in the bone marrow. A key transcription factor essential for the early development of B cells in the bone marrow is a cellular myelocytomatosis (c-Myc) protein. c-Myc is one of the most critical transcription factors, regulating various cellular processes such as cell proliferation, cell growth, and cell apoptosis [4]. The *c-Myc* proto-oncogene, located at chromosome 8q24, is frequently involved in chromosomal translocation—most notably t(8;14)(q24;q32)—which result in aberrant c-Myc expression, a common feature in human cancer pathogenesis. Dysregulation of c-Myc in B cell lymphocytes can lead to Burkitt’s lymphoma, a highly aggressive disorder characterized by an exceptionally rapid proliferation of B cell proliferation. In all cases of Burkitt’s lymphoma, the *c-Myc* gene is translocated to one of the immunoglobulin loci enhancers, resulting in markedly elevated c-Myc mRNA and protein expression. Key signaling molecules involved in c-Myc activation include PI3K, AKT, and mTOR [5].

Currently, chemotherapy remains the most common approach for treating lymphoma. In addition, bone marrow transplantation serves as a secondary treatment option for lymphoma patients. Most chemotherapeutic agents function by inhibiting DNA synthesis in cancer cells; however, they also affect normal cells undergoing DNA replication, leading to damage in both malignant and healthy cells. These drugs are often associated with adverse effects, including hair loss, nausea, vomiting, and diarrhea [6]. To address these limitations, several studies have explored alternative therapies for Burkitt’s lymphoma, including natural products. For example, shikonin has been shown to suppress cell proliferation and induce caspase-dependent apoptosis in Burkitt’s lymphoma cells by targeting the c-Myc and PI3K/AKT/mTOR pathways [7]. Curcuminoids have also emerged as promising candidates for cancer treatment due to their broad biological activities, low toxicity, and potential benefits for immune-related, metabolic diseases [8]. Previous studies have demonstrated that curcumin can inhibit tumor growth in Burkitt’s lymphoma Raji xenograft mice models [9]. However, its effects on *c-Myc* gene expression in Raji cells have not been thoroughly investigated. Plants from Zingiberaceae family, widely used in traditional medicine and Thai cuisine, are of particular interest for their potential anti-lymphoma properties, including the ability to inhibit cell proliferation and induce apoptosis.

This study began by examining the cytotoxicity of plant extracts and major pure compounds derived from members of the Zingiberaceae family. Five species will be selected for investigation including galangal (*Alpinia galanga*), black turmeric (*Curcuma aeroginosa*), black ginger (*Kaempferia parviflora*), phlai lueang (*Zingiber montanum*), and phlai dam (*Zingiber ottensii*). The subsequent phase focused on elucidating the inhibitory mechanisms of the most potent plant extract, particularly its effects on cell proliferation and apoptosis induction in lymphoma cell lines. In the context of blood cancers, previous studies have investigated crude extracts and isolated compounds from the Zingiberaceae plants in leukemic cells. For instance, curcumin from turmeric has been shown to significantly reduce *WT1* gene expression, leading to decreased WT1 mRNA levels in patient-derived leukemic cells [10,11]. Pure curcumin has been found to suppress *WT1* gene expression in K562 cells via protein kinase C alpha (PKCα) signaling, upstream of WT1 auto-regulatory transcriptional loop [12]. Among curcuminoids, bisdemethoxycurcumin has demonstrated strong inhibitory effects on the KG-1a leukemic stem cell line through WT1 suppression [13]. Additionally, 6-Gingerol from ginger has been reported to inhibit myeloid cell proliferation and induce apoptosis through generation of reactive oxygen species (ROS) [14,15]. Galangals A and B, isolated from Japanese ginger flower buds, exhibited strong cytotoxicity against human T lymphoma Jurkat cells [16]. However, evidence remains limited regarding the effects of Zingiberaceae-derived crude extracts or pure compounds on c-Myc protein expression and related signaling pathways in lymphoma cells. Notably, zerumbone, a natural compound derived from *Zingiber zerumbet*, has been shown to inhibit Raji cell proliferation by modulating apoptosis-related gene expressions, suggesting its potential as a therapeutic candidate for Burkitt’s lymphoma [17].

Currently, no cancer medication has been specifically developed to selectively target cancer cells without affecting normal cells. In this study, five Zingiberaceae plants were studied for cytotoxic effects against lymphoma cell lines (Raji and Daudi), in comparison to normal peripheral blood mononuclear cells (PBMCs). Following the initial screening, the selected plant extract and its major pure compounds were investigated for their inhibitory effects on c-Myc protein expression, which is known to regulate cell proliferation in lymphoma cells. In addition, their effects on the cell cycle and apoptosis were examined to support their antiproliferative activities. The outcomes of this study are expected to provide novel insights for researchers and herbal products developers, contributing to the advancement of plant-based therapeutics for lymphoma treatment in future.

## 2. Materials and Methods

### 2.1. Crude Extracts of Five Zingiberaceae Plants

In June 2019, fresh rhizomes of *Alpinia galanga*, *Curcuma aeroginosa, Kaempferia parviflora*, *Zingiber montanum*, and *Zingiber ottensii* were collected from a local garden in Chiang Mai, Thailand. These plants are common medicinal species in Thailand and are widely used in Thai disc recipes and traditional medicine. They are not considered at risk of extinction. All plant materials were identified and authenticated by Wannaree Charoensup of Chiang Mai University, Chiang Mai, Thailand. Voucher specimens (deposition no. 009245 for *A. galanga*, 0023261 for *C. aeroginosa*, 0023370 for *Kaempferia parviflora,* 004581 for *Z. montanum*, 000109 for *Z. ottensii*) were deposited in the Herbarium, Faculty of Pharmacy, Chiang Mai University, Thailand.

Ethanolic extracts of these fresh rhizomes were obtained using the maceration technique. Briefly, each rhizome was dried and ground into a powder, which was then macerated with 95% ethanol at a ratio of 1:3 (*w*/*v*) for 48 h. The filtrates were collected using a filter cloth, and the residue was subjected to three additional rounds of maceration under the same condition. The combined filtrates were evaporated to dryness using a rotary evaporator. The resulting extracts were stored in light-resistant containers at −20 °C. Standard gingerol and 1′-acetoxychavicol acetate (ACA) were purchased from Sigma-Aldrich (St. Louis, MO, USA).

### 2.2. Cell Culture Condition for Lymphoma Cell Lines

Raji and Daudi are both human B cell lymphoblastoid cell lines derived from patients with Burkitt’s lymphoma and characterized as suspension cell lines. Raji harbors a t(8;14) translocation, resulting in the *MYC-IGH* fusion gene, and contains latent Epstein–Barr virus (EBV) in a non-producer state. In contrast, Daudi carries the full EBV genome and expresses mRNA for the proto-oncogene *BCL2*. Raji (EP-CL-0189) was purchased from Elabscience^®^, Houston, TX, USA. Daudi (CCL-213) was obtained from ATCC, Manassas, VA, USA. Raji and Daudi cells were cultured in RPMI-1640 medium (Roswell Park Memorial Institute, Invitrogen™, CA, USA) supplemented with 10% fetal bovine serum, 2 mM L-glutamine, 100 units/mL penicillin, and 100 μg/mL streptomycin (Invitrogen™, Carsbad, CA, USA). These two cell lines were maintained at 37 °C in humidified incubators with 5% CO_2_.

### 2.3. Total Cell Number Count by Trypan Blue Exclusion Test

Live cells have intact membranes and can exclude trypan blue dye, whereas dead cells with compromised membranes were stained by the dye. A cell suspension with 0.2% trypan blue solution, and viable (unstained) and dead (stained) cells were counted using a hemacytometer. The percentage of viable cells was then calculated.

### 2.4. Cytotoxicity of Crude Extracts and Candidate Main Pure Compound from Zingiberacaeae on Lymphoma Cell Lines

The cytotoxicity screening of Zingiberaceae family extracts against Raji and Daudi cells was performed using MTT assay (3-(4,5-dimethythiazol-2-thizolyl)-2,5-diphenyl tetrazolium bromide) in three independent experiments. Raji and Daudi cells were adjusted to 1.0 × 10^4^ cells/mL in RPMI-1640 complete medium supplemented with 10% FBS.

The cells were harvested and washed three times with PBS, pH 7.4. Based on their instinct growth rates (Appendix A), Raji and Daudi cells were seeded at different initial densities—1.0 × 10^5^ cells /mL for Raji cells and 2.0 × 10^5^ cells /mL for Daudi cells—into 96-well plates, followed by incubation at 37 °C in a humidified atmosphere containing 5% CO_2_ for 24 h. After incubation, cells were treated with crude extracts from Zingiberacaeae plants at concentrations ranging from 3.125 to 100 µg/mL and cultured in complete medium for 48 h. Subsequently, 15 µL of MTT dye solution (5.0 g/L) was added to each well, followed by incubation for 4 h. Formazan crystals formed by viable cells were then dissolved in DMSO, and absorbance was measured at 578 nm using microplate reader (Metertech, Nankang, Taipei, Taiwan), with 630 nm as the reference wavelength. The percentage of surviving cells was calculated using the following equation:(1)$$\begin{array}{cc} \%\;\mathrm{Cell}\;\mathrm{viability}\;= & \frac{\mathrm{Mean}\;\mathrm{absorbance}\;\mathrm{in}\;\mathrm{test}\;\mathrm{well}}{\mathrm{Mean}\;\mathrm{absorbance}\;\mathrm{in}\;\mathrm{vehicle}\;\mathrm{control}\;\mathrm{well}} \end{array}\;100$$

The average percentage of cell viability at each concentration, obtained from triplicate experiments, was plotted as a dose–response curve. The inhibitory concentrations at 20% (IC_20_) and 50% (IC_50_) of each crude extract were determined as the lowest concentrations that inhibited cell growth by 20% or 50% in treated cells compared to untreated controls. The IC_20_ and IC_50_ values of each Zingiberaceae plant extract were calculated and used to select suitable concentrations of crude extracts for further study. Candidate crude extracts and their major pure compounds showing low concentrations IC_50_ values (indicating good activity) were selected for the MTT assays in Raji and Daudi cells, with comparisons made to normal peripheral blood mononuclear cells (PBMCs). DMSO (0.8%) was used as a vehicle control.

The selectivity index (SI) of active compounds and chemotherapeutic drugs was calculated using the following equation.(2)$$\mathrm{Selectivity}\;\mathrm{index}\;(\mathrm{SI})=\frac{{\mathrm{IC}}_{50\;N}}{{\mathrm{IC}}_{50\;C}}$$
where IC_50N_ is IC_50_ for normal cells (PBMCs) and IC_50C_ is IC_50_ for cancer cells which were treated with the same compounds in both cells.

### 2.5. Cytotoxicity of Pure Compounds from Zingiberaceae on PBMCs

#### 2.5.1. PBMC Preparation by Ficoll-Hypaque Density Gradient Centrifugation Method

Blood samples (10–20 mL) were collected in heparin tubes from at least 5 healthy volunteers. The use of human PBMCs in this study was approved by the Human Research Ethics Unit of the Faculty of Associated Medical Sciences, Chiang Mai University (Approval No. 307/2024; date of approval: 3 July 2024). Written informed consent was obtained from all participants, who were fully informed of the study’s objectives and procedures. Participants were assured that their personal information and all procedures were conducted in accordance with relevant guidelines and regulations. Then, collected blood samples were diluted with sterile PBS, pH 7.4, at a 1:1 (*v*/*v*) ratio to reduce viscosity. The diluted blood was gently overlaid onto the Ficoll-Hypaque and centrifuged at 400× *g* for 30 min at room temperature without a break, allowing for separation into plasma, PBMC, Ficoll-Hypaque, and RBC layers. The PBMC layer was carefully harvested and transferred to a new tube. Sterile PBS, pH 7.4 was added, followed by gentle mixing and centrifugation at 2000 rpm for 10 min at room temperature. The supernatant was removed, and the washing and centrifugation steps were repeated. Cell numbers were determined using the trypan blue exclusion method.

#### 2.5.2. Cytotoxicity Effect of Major Pure Compound of Candidate Plant

PBMCs were isolated from whole blood from Section 2.5.1. The cells were adjusted to 1.0 × 10^7^ cells/mL in RPMI-1640 complete medium supplemented with 10% FBS and seeded into 96-well plates. The plates were incubated in a CO_2_ incubator at 37 °C with 5% CO_2_ for 24 h. Subsequently, the cells were treated with the major pure compound from the candidate Zingiberaceae plant at concentrations ranging from 3.125 to 100 µg/mL and cultured in complete medium for 48 h. After treatment, 15 µL of MTT dye solution was added to each well and incubated for 4 h. Finally, the formazan crystals formed were dissolved with DMSO, and absorbance was measured at 578 nm using a microplate reader, with 360 nm as the reference wavelength.

### 2.6. Investigation of Candidate Zingiberaceae Plants and Major Pure Compounds on Cell Cycle Arrest in Lymphoma Cell Lines

Raji and Daudi cells were adjusted to 1.0 × 10^5^ and 2.0 × 10^5^ cells/mL, respectively, in RPMI-1640 complete medium supplemented with 0.5% FBS. The cells underwent overnight incubation at 37 °C in a humidified atmosphere with 5% CO_2_ to induce cell starvation. After starvation, the cells were treated with candidate Zingiberaceae extracts and their major pure compounds at concentrations corresponding to their IC_20_ values and incubated under the same conditions for 48 h. Doxorubicin was used as a positive control and treated under similar conditions. Following treatment, the cells were harvested and washed three times with cold PBS, pH 7.4, then transferred into microcentrifuge tubes. The cells were fixed by adding ice-cold 70% ethanol in PBS and incubated at 4 °C in the dark for 30 min. After fixation, the cells were centrifuged, the supernatant was discarded, and the cell pellets were stained with a working propidium iodide (PI) solution containing PI, EDTA, Triton X, and RNase A. The fluorescence signals were measured using a flow cytometer, and the data were analyzed using FlowJo V10 software.

### 2.7. Investigation of c-Myc Protein Expression by Western Blotting

To investigate the protein expression after treatment with candidate major pure compounds for 48 h, Western blotting was performed. After 48 h of treatment, both untreated and treated cells were lysed using cell lysis buffer containing protease inhibitors. Protein concentrations were determined using the Pierce™ BCA Protein Assay Kit (ThermoFisher Scientific, Waltham, MA, USA). Equal amounts of protein (30 µg/sample) were separated on 12% sodium dodecyl sulfate-polyacrylamide gel (SDS-PAGE) and transferred onto PVDF membranes. Membranes were blocked with 5% skim milk in PBS, pH 7.4, for 2 h at room temperature, followed by incubation with primary antibody: rabbit monoclonal anti-c-Myc IgG and anti-p-c-Myc (Ser62)(E1J4K) (Cell Signaling Technology, Danvers, MA, USA), at a dilution of 1:1000, and rabbit polyclonal anti-human GAPDH IgG (Santa Cruz Biotechnology, Santa Cruz, CA, USA) at 1:16,000 dilution. In this study, the membrane initially probed with an anti-c-Mcy antibody was stripped using Restore PLUS Western Blot Stripping Buffer (ThermoFisher Scientific, Waltham, MA, USA), and subsequently reprobed with an anti-p-c-Myc antibody. Both c-Myc and p-c-Myc protein have a molecular weight of approximately 62 kDa. Incubation was carried out with gentle shaking for 2 h. Membranes were then washed and incubated with HRP-conjugated goat anti-rabbit IgG (Promega, Madison, WI, USA) at 1:20,000 dilution for 1 h with shaking. Protein bands were visualized using Luminata™ Forte Western HRP substrate (Merck, Darmstadt, Germany) and exposed to X-ray film (Sakura, Osaka, Japan) or detected by ChemiDoc MP Imaging Systems (Bio-Rad, Hercules, CA, USA). Densitometric analysis was performed using Quantity One 1-D Analysis software version 4.6.6 (Bio-Rad, Hercules, CA, USA). The intensities of c-Myc and p-c-Myc bands were normalized to GAPDH.

### 2.8. Bioinformatics Analysis

To investigate the biological significance of c-Myc in lymphoma, KEGG pathway enrichment and Gene Ontology (GO) term analysis were performed [18]. Gene enrichment data were retrieved from GeneCards (www.genecards.org) and validated using DAVID (https://david.ncifcrf.gov/, accessed on 24 October 2024) [19,20]. KEGG pathways associated with lymphoma were identified based on their involvement in cancer-related signaling, immune regulation, and apoptosis. GO analysis was conducted to classify lymphoma-associated genes into three categories: Biological Process (BP), Cellular Component (CC), and Molecular Function (MF). The results were visualized using SRplot (https://www.bioinformatics.com.cn/srplot, accessed on 24 October 2024), highlighting significantly enriched terms based on *p*-values (-log10 transformed) [21].

To further explore the role of c-Myc in lymphoma, protein–protein interaction (PPI) network analysis was conducted using STRING v12.0 (https://string-db.org/, accessed on 24 October 2024) [22]. The resulting network data were imported into Cytoscape v3.10.3 for visualization and network topology analysis [23]. Key interacting proteins were identified, emphasizing those with critical regulatory functions in lymphoma, including *XPO1*, *BCL2*, *TP53*, and other genes.

### 2.9. Investigation of Candidate Zingiberaceae Plants and Major Pure Compounds on Apoptosis Induction in Lymphoma Cell Lines

Apoptosis induction by candidate Zingiberaceae plants and their major pure compounds were determined using flow cytometry. The Annexin V-FITC Apoptosis Detection Kit^®^ (BD bioscience, Franklin Lakes, NJ, USA) was used in this experiment. Raji and Daudi cells were adjusted to 1.0 × 10^5^ and 2.0 × 10^5^ cells/mL, respectively, in RPMI-1640 complete medium supplemented with 10% FBS and treated with the candidate plant extracts and their major pure compounds at concentrations ranging from their IC_20_ to IC_50_ values. The cells were incubated at 37 °C in a humidified atmosphere containing 5% CO_2_ for 48 h. Doxorubicin was used as a positive control under the same conditions. After treatment, the cells were harvested, washed three times with cold PBS at pH 7.4, and counted. The cells were then resuspended in 1× Annexin V binding buffer at the concentration of 1 × 10^6^ cells/mL. A volume containing 1 × 10^5^ cells was transferred to a microcentrifuge tube, followed by the addition 5 µL of Annexin V-FITC and propidium iodide (PI). The cells were gently vortexed and incubated for 15 min at 25 °C in the dark. Subsequently, 1× binding buffer was added to each tube, and the samples were analyzed by flow cytometer within 1 h. Data analysis was performed using FlowJo V10 software. Unstained cells and single-stained cells (Annexin V-FITC or PI only) were used to set the quadrants for distinguishing cell populations. The percentage of apoptotic cells was calculated by subtracting the percentage of apoptotic cells in untreated control from those in treated groups.

### 2.10. Statistical Analysis

Data were expressed as the mean ± standard deviation (SD) or the mean ± standard error of the mean (SEM) from triplicate samples of three independent experiments. Statistical differences between the means were determined using one-way ANOVA and Student’s *t*-test. Differences were considered statistically significant when the *p* value was less than 0.05 (*p* < 0.05), 0.01 (*p* < 0.01), and 0.001 (*p* < 0.001).

## 3. Results

### 3.1. Cytotoxicity of Crude Extracts from Zingiberacaeae on Lymphoma Cell Lines

In this study, the crude ethanolic extracts of five Zingiberaceae plants were examined, and their cytotoxic effects were compared. The results indicated that each crude extract exhibited varying levels of cytotoxicity against Raji and Daudi cell lines. Among them, galangal (*Alpinia galanga*) was the most effective in both Raji and Daudi cells, particularly in Daudi cells, with IC_50_ values of 31.52 ± 3.59 and 14.20 ± 2.34 µg/mL, respectively, (Table 1). The Selectivity Index (SI), defined as the ratio between cytotoxicity in normal cells and anticancer efficacy, serves as an indicator of compound’s therapeutic window. A higher SI value implies a more favorable safety profile, reflecting greater selectivity towards malignant cells over healthy ones [24,25,26]. In this study, all crude extracts showed selectivity index values greater than 1, indicating higher specificity towards lymphoma cells than normal PBMCs (Table 1). Consequently, galangal was selected for further investigation. Based on previous literature [27,28,29], galangin and 1′-acetoxychavicol acetate (ACA) were identified as the major active constituents of galangal and were further examined for their biological activities in this study.

### 3.2. Cytotoxicity of Major Compounds from Galangal on Lymphoma Cell Lines (Raji and Daudi) and Normal PBMCs

Based on the superior cytotoxicity of galangal extract observed in the previous experiment, its two major active compounds, galangin and ACA, were further evaluated. The cytotoxic effects of these compounds were evaluated against Raji and Daudi lymphoma cell lines and compared with those on normal PBMCs. The IC_50_ values of galangin and ACA in Raji cells were 21.00 ± 1.58 and 1.93 ± 0.26 µg/mL, respectively. Similarly, in Daudi cells, the IC_50_ values were 10.75 ± 1.29 and 1.74 ± 0.46 µg/mL, respectively. These results indicated that ACA exerted the most potent cytotoxicity against both lymphoma cell lines. However, it also demonstrated cytotoxic effects on PBMCs (IC_50_ = 4.69 ± 0.25 µg/mL). In contrast, galangin showed much lower cytotoxicity toward PBMCs (IC_50_ > 100 µg/mL), as shown in Table 2. Notably, the concentration of ACA should be maintained below 4.69 µg/mL to avoid cytotoxic effects on normal cells.

In the present study, the SI values for galangin, ACA, and doxorubicin were greater than 1, indicating greater specificity towards lymphoma cells than toward normal PBMCs (Table 2).

### 3.3. Effect of Galangin and ACA on Cell Cycle in Raji and Daudi Cells

Because the MTT assay based on cytotoxicity assessment could not distinguish whether the observed reduction in cell viability resulted from suppressed cell proliferation or cell death, cell cycle analysis was subsequently conducted to clarify the underlying mechanism. To investigate the effect of galangin and ACA on cell cycle progression, Raji and Daudi cells were treated with each compound at the IC_20_ concentration, as determined from prior MTT cytotoxicity assays. After 48 h of treatment, the distribution of cell across the G0/G1, S, and G2/M phases was analyzed using propidium iodide (PI) staining and flow cytometry. In Raji cells, galangin treatment significantly arrested the cell cycle at the S phase (*p* < 0.05) compared with the vehicle control (Figure 1a). In contrast, in Daudi cells, galangin treatment significantly arrested the cell cycle at the G0/G1 phase (*p* < 0.05) compared with the vehicle control (Figure 1b). Nevertheless, ACA treatment significantly increased the proportion of cells in the sub-G1 phase, indicating apoptosis, in both cell lines. Doxorubicin (positive control) significantly arrested the cell cycle at the G2/M phase in Raji cells (Figure 1a) and at G0/G1 phase in Daudi cells (Figure 1b).

### 3.4. Effects of Galangin and ACA on c-Myc and Phosphorylated c-Myc Protein Expressions in Raji and Daudi Cells

The c-Myc protein is a critical transcription factor that regulates several cellular processes, including cell proliferation, growth, and apoptosis in lymphoma cells. It plays a central role in cell cycle progression and also serves as a biomarker in lymphoma cells [30]. In this study, c-Myc and its phosphorylated (activated) form, p-c-Myc, were evaluated as biomarkers of cell proliferation. Raji and Daudi cells were treated with galangin or ACA at concentrations corresponding to IC_20_, IC_30_, and IC_50_ for 48 h. Protein expression levels of c-Myc and p-c-Myc were determined by Western blot analysis. Notably, in Daudi cells, extending the treatment to 48 h resulted in excessive cell death and an insufficient number of viable cells for analysis as shown in Appendix A. Therefore, these cells were analyzed after 24 h of treatment. The result demonstrated that both galangin and ACA decreased c-Myc and p-c-Myc protein levels in a dose-dependent manner in both Raji and Daudi cells. Specifically, ACA suppressed c-Myc protein expression in Raji cells by 18.75, 24.41, and 48.86% at IC_20_, IC_30_, and IC_50_ concentrations, respectively, after 48 h of treatment. In addition, p-c-Myc expression was reduced by 9.18, 33.56, and 35.12%, respectively, compared with the vehicle control (Figure 2a,b). In Daudi cells, ACA treatment produced a more pronounced suppression, reducing c-Myc protein expression by 37.58, 62.73, and 71.68% at IC_20_, IC_30_, and IC_50_, respectively, after 24 h. Similarly, p-c-Myc expression decreased by 30.32, 50.77, and 59.84%, respectively, compared with the vehicle control (Figure 2d,e). Galangin also significantly downregulated c-Myc and p-c-Myc expression, though to a slightly lesser extent than ACA. In Raji cells, galangin reduced c-Myc levels by 15.57, 26.25, and 41.33% at IC_20_, IC_30_, and IC_50_, concentrations, respectively, after 48 h of treatment, while p-c-Myc expression was reduced by 7.44, 18.96, and 34.60%, respectively, compared with the vehicle control (Figure 2a,b). In Daudi cells, galangin decreased c-Myc protein expression by 37.80, 49.82, and 64.63%, and reduced p-c-Myc expression by 27.34, 52.37, and 66.77% at IC_20_, IC_30_, and IC_50_, respectively, after 24 h of treatment (Figure 2d,e). Interestingly, these results suggest that both galangin and ACA suppress c-Myc signaling in lymphoma cells in a dose-dependent manner, with ACA exhibiting a more pronounced inhibitory effect.

### 3.5. Effect of Galangin and ACA on Total Cell Number in Raji and Daudi Cells

As shown Figure 2c,f, galangin and ACA significantly decreased the total cell number, accompanied by a corresponding increase in the percentage of cell death. Raji and Daudi cells were treated with IC_20_, IC_30_, and IC_50_ concentrations of galangin or ACA for 48 h in Raji cells and 24 h in Daudi cells. Cell viability was assessed using the trypan blue exclusion assay. Notably, ACA reduced the total number of viable cells by approximately 40% in Raji cells and 60% in Daudi cells. In parallel, ACA significantly increased the proportion of non-viable (trypan blue-positive) cells, with maximum increases of 23 and 40% in Raji and Daudi cells, respectively. In contrast, galangin reduced the total cell numbers by approximately 50% in Raji cells and 30% in Daudi cells in a dose-dependent manner but did not significantly increase cell death in either cell line. These findings indicated that ACA not only inhibited cell proliferation more effectively than galangin but also actively promoted cell death, highlighting its potential as a more potent anti-lymphoma agent.

**Figure 2 biology-14-01098-f002:**
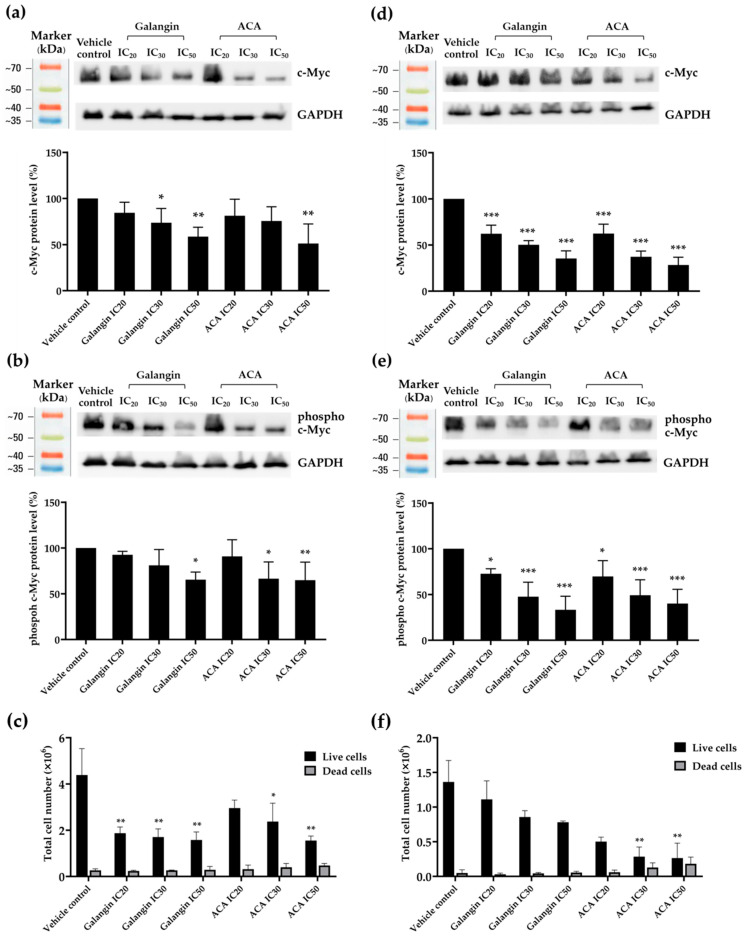
Effects of galangin and ACA on c-Myc and p-c-Myc protein expression and cell numbers in Raji and Daudi cells, assessed by Western blotting and the trypan blue exclusion assay. (**a**) c-Myc protein levels in Raji cells following treatments with galangin and ACA at IC_20_, IC_30_, and IC_50_ concentrations for 48 h. (**b**) p-c-Myc levels in Raji cells under the same treatment and analysis conditions as in (**a**). (**c**) Total cell numbers in Raji cells after 48 h treatment with galangin and ACA, determined using the trypan blue exclusion method. (**d**) c-Myc protein levels in Daudi cells treated with galangin and ACA at IC_20_, IC_30_, and IC_50_ concentrations for 24 h, assessed via Western blotting and densitometry. (**e**) P-c-Myc levels in Daudi cells under the same treatment and analysis conditions as in (**d**). (**f**) Total cell numbers in Daudi cells following 24 h treatment, determined using the trypan blue exclusion assay. Each bar represents the mean ± SD of three independent experiments. Protein expression was analyzed by Western blotting and quantified using a scanning densitometer. GAPDH served as a normalization control. C-Myc and p-c-Myc levels were normalized to GAPDH as loading control. Asterisks (*) indicate statistically significant differences compared to the vehicle control (* *p* < 0.05, ** *p* < 0.01, *** *p* < 0.001).

### 3.6. Network Analysis of Galangin and ACA for Predicting Targets Against Lymphoma

Using PharmMapper and SwissTargetPrediction, 358 and 347 candidate gene targets were identified for galangin and ACA, respectively. Lymphoma-related genes were predicted using two online human gene databases, DisGeNET and GeneCards, yielding 10,038 candidate gene targets. All collected gene targets were classified into three groups: (1) genes related to lymphoma, (2) genes targeted by galangin or ACA, and (3) overlapping genes between lymphoma and either galangin or ACA. As shown in Figure 3a and Figure 4a, 280 gene targets overlapped between lymphoma and galangin, while 267 overlapped between lymphoma and ACA. To further explore the mechanisms of galangin and ACA in lymphoma, protein–protein interaction (PPI) networks were constructed using the STRING database and visualized in Cytoscape. Based on node degree values, the top 20 key gene targets were identified (Figure 3b and Figure 4b). Functional enrichment analysis of the 280 overlapping galangin-lymphoma gene targets, performed using DAVID, revealed significant enrichment in 490 biological processes (BP), 65 cellular components (CC), and 136 molecular functions (MF) (*p* < 0.05) (Figure 3c). Galangin may regulate phosphorylation, signal transduction, and apoptosis within the cytosol, cytoplasm, nucleus, and plasma membrane, through activities such as protein binding, ATP binding, and metal ion binding, thereby inhibiting lymphoma progression. KEGG pathway analysis showed that these gene targets were involved in several important pathways, including metabolic pathways, pathways in cancer, the PI3K-Akt signaling pathway, the MAPK signaling pathway, and apoptosis (Figure 3d). Similarly, the 267 overlapping, ACA—lymphoma bene targets were enriched in 478 biological processes, 75 cellular components, and 152 molecular functions (*p* < 0.05) (Figure 4c). Gene ontology analysis suggested that ACA may regulate cellular mechanisms in a manner similar to galangin, contributing to the inhibition of lymphoma progression. KEGG pathway analysis of ACA-associated gene targets also revealed involvement in metabolic pathways, pathways in cancer, the PI3K-Akt signaling pathway, the MAPK signaling pathway, and apoptosis (Figure 4d). Therefore, both galangin and ACA may inhibit the PI3K-Akt signaling pathway through their action on Akt, thereby contributing to their anti-lymphoma activity.

### 3.7. Effects of Galangin and ACA on Cleaved-Casp3 Expressions in Raji and Daudi Cells by Western Blotting Assay

Cleaved caspase-3 (cl-Casp3) is a critical executioner protease in the apoptotic cascade, responsible for orchestrating the dismantling of cellular structures during programmed cell death. Its activation serves as a key hallmark of apoptosis. In the study, the pro-apoptotic effects of galangin and ACA were evaluated in Raji and Daudi lymphoma cell lines by assessing cl-Casp3 protein levels following treatment. Cells were treated with IC_20_, IC_30_, and IC_50_ concentrations of galangin or ACA for 48 h in Raji cells and 24 h in Daudi cells. Western blot analysis revealed a dose-dependent increase in cl-Casp3 expression in both cell lines following treatment with either compound. ACA significantly increased cl-Casp3 expression by 9.86-fold in Raji cell and 3.42-fold in Daudi cells after treatment at IC_50_ concentration compared with the vehicle control (Figure 5a,c). Notably, ACA induced a more pronounced upregulation of cl-Casp3 than galangin, consistent with its stronger effect on promoting cell death. These findings confirmed that both galangin and ACA activated the intrinsic apoptotic pathway in lymphoma cells, with ACA exhibiting greater potency.

### 3.8. Effects of Galangin and ACA on Apoptosis in Raji and Daudi Cells by Flow Cytometric Analysis

To further validate the pro-apoptotic effects of galangin and ACA, Annexin V-FITC/propidium iodide (PI) dual staining was performed, followed by flow cytometric analysis. This assay distinguishes early apoptotic cells (Annexin V^+^/PI^−^), late apoptotic or necrotic cells (Annexin V^+^/PI^+^), and viable cells (Annexin V^−^/PI^−^). Raji (1.0 × 10^5^ cells/mL) and Daudi cells (2.0 × 10^5^ cells/mL) were treated with IC_20_, IC_30_, and IC_50_ concentrations of galangin or ACA for 48 h; doxorubicin was used as the positive control. The results demonstrated a dose-dependent increase in the percentage of apoptotic cells in both cell lines after treatment with either compound. ACA significantly induced apoptosis, reaching 46.03 ± 4.10% in Raji cells and 24.53 ± 6.86% in Daudi cells at the IC_50_ concentration (Figure 5b,d). In Raji cells, galangin significantly induced apoptosis at the IC_50_ concentration; however, the percentage was relatively low (7.88 ± 3.77%). In contrast, in Daudi cells, galangin induced higher levels of apoptosis, with rates of 20.29 ± 2.82, 25.92 ± 1.88, and 31.29 ± 3.68% at IC_20_, IC_30_, and IC_50_ concentrations, respectively, after 48 h of treatment (Figure 5d). ACA markedly promoted apoptosis, with a significant accumulation of both early and late apoptotic populations at IC_50_ in both cell lines. Galangin also promoted apoptosis, albeit to a lesser extent than ACA in Daudi cells. Importantly, the increase in cl-Casp3 expression directly correlated with the proportion of apoptotic cells, confirming its role as a molecular marker of cell death triggered by galangin and ACA. These findings corroborated the biochemical data from cl-Casp3 analysis and further confirmed that both compounds effectively induced apoptosis in lymphoma cells, with ACA demonstrating superior potency.

**Figure 5 biology-14-01098-f005:**
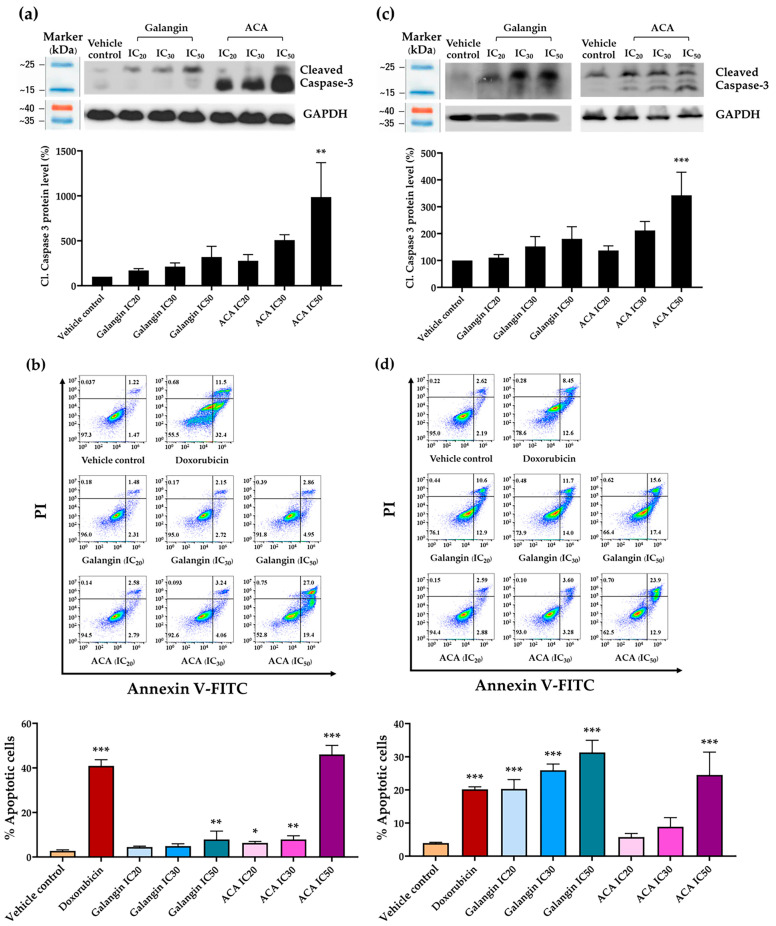
Effects of galangin and ACA cell cycle in Raji and Daudi cells. (**a**) Cleaved caspase-3 (cl-casp3) protein levels in Raji cells (1.0 × 10^5^ cells/mL) following treatment with galangin and ACA at IC_20_, IC_30_, and IC_50_ concentrations for 48 h. Protein expression was assessed by Western blotting and quantified using a scanning densitometer. Cl-Casp3 levels were normalized to GAPDH as a loading control. (**b**) Raji cells (1.0 × 10^5^ cells/mL) were treated with galangin and ACA at IC_20_, IC_30_, and IC_50_ concentrations for 48 h. Apoptotic cells were identified by Annexin V-FITC and PI staining. Representative flow cytometry dot plots display the distribution of cell populations (top). (**c**) Cl-Casp3 protein levels in Daudi cells (2.0 × 10^5^ cells/mL) after 24 h treatment with galangin and ACA at IC_20_, IC_30_, and IC_50_ concentrations, analyzed and quantified as described in (**a**). (**d**) Daudi cells (2 × 10^5^ cells/mL) were treated with galangin and ACA at IC_20_, IC_30_, and IC_50_ concentrations for 48 h. Apoptosis was assessed by Annexin V-FITC/PI staining, and representative flow cytometry plots illustrate the distribution of apoptotic populations. Each bar represents the mean ± SD of three independent experiments. Asterisks (*) indicate statistically significant differences compared to the vehicle control (* *p* < 0.05, ** *p* < 0.01, *** *p* < 0.001).

## 4. Discussion

Members of Zingiberaceae family, long valued in traditional medicine, have recently gained attention for their potential to mitigate chemotherapy-related side effects and improve lymphoma treatment outcomes. Previously studies have shown that compounds such as curcumin and zerumbone exhibited anti-lymphoma activity via NF-κB inhibition and mitochondrial-mediated apoptosis through upregulation of pro-apoptotic proteins and suppression of c-Myc in Burkitt’s lymphoma cells [8,9]. Additionally, zerumbone—a sesquiterpene extracted from wild ginger (*Zingiber zerumbet*)—has been reported to inhibit Raji cell proliferation by inducing late apoptosis and modulating apoptosis-related genes, including Bax upregulation and Bcl-2 and c-Myc downregulation, underscoring its therapeutic potential against Burkitt’s lymphoma [17]. In the present study, the cytotoxic effects of five Zingiberaceae plants, including galangal (*Alpinia galanga*), black turmeric (*Curcuma aeroginosa*), black ginger (*Kaempferia parviflora*), phlai lueang (*Zingiber montanum*) and phlai dam (*Zingiber ottensii*), were evaluated in two lymphoma cell lines (Raji and Daudi). All crude extracts exhibited cytotoxicity to varying degrees, with galangal showing the strongest effects (IC_50_ = 31.52 ± 3.59 µg/mL in Raji; 14.20 ± 2.34 µg/mL in Daudi cells). Based on these findings, galangal and its major pure compound were selected for further investigation.

Galangal, a widely used plant in traditional Thai and Indonesian medicine and cuisine, contains two prominent bioactive compounds: galangin and 1′-acetoxychavicol acetate (ACA). Galangin, a flavonol-type flavonoid, possesses antioxidant, antibacterial, antiviral, and anti-proliferative properties, and has been shown to inhibit breast cancer cell growth by modulating oxidative stress pathways [31,32]. ACA, a phenylpropanoid ester primarily isolated from *A. galanga* and *A. conchigera*, demonstrates potent anticancer, anti-inflammatory, and NF-κB inhibitory activities. It promotes apoptosis in various cancer cell lines and is considered a promising candidate for targeted drug delivery due to its hydrophobicity and selective cytotoxic effects [33,34].

In this study, ACA exhibited significantly higher cytotoxicity against Raji and Daudi lymphoma cells compared with galangin, although ACA also displayed notable toxicity toward normal PBMCs, indicating a narrower therapeutic window. This observation aligns with previous reports of ACA’s potent anticancer activity coupled with low toxicity towards normal cells, such as its selective inhibition of triple-negative breast cancer cells (IC_50_ < 30 µM) while sparing human mammary epithelial cells (HMECs) [35]. Similarly, ACA has been shown to induce apoptosis in myeloma and leukemic models via mitochondrial and dead receptor pathways without significant toxicity to healthy tissue [36]. By contrast, galangin has been demonstrated to have strong anti-tumor effects with minimal off-target toxicity. For instance, galangin suppresses PD-L1 expression, enhances T cell-mediated tumor cytotoxicity, and inhibits hepatocellular carcinoma growth in vivo [37]. The higher IC_50_ values of galangin for lymphoma cells (21.00 ± 1.58 µg/mL in Raji; 10.75 ± 1.29 µg/mL in Daudi) compared with ACA suggest a more favorable safety profile in normal cells. Cell cycle analysis revealed that galangin at IC_20_ for 48 h induced phase-specific arrest, which differed between cell lines—S phase in Raji and G0/G1phase in Daudi—likely due to phenotypic differences. Raji cells harbor a mutated p53, which can impair the DNA damage response and lead to S phase arrest due to checkpoint dysfunction, whereas Daudi cells express wild-type p53, enabling canonical G0/G1 arrest in response to stress signals [38]. In contrast, ACA treatment resulted in significant sub-G1 accumulation in both cell lines, indicating cell death without detectable cell cycle arrest.

Both galangin and ACA inhibited c-Myc and p-c-Myc expression in a dose-dependent manner, with Daudi cells showing >70% c-Myc and ~60% p-c-Myc reduction at IC_50_. These results align with earlier studies showing galangin’s ability to suppress PD-L1 via STAT3 and c-Myc inhibition [37] and ACA’s capacity to inhibit proliferation and induce apoptosis via HER2/MAPK/ERK and PI3K/Akt signaling pathways as well as NF-κB suppression in multiple myeloma cells [39] and promoting DNA fragmentation in NB4 cells [36]. Functionally, ACA not only reduced total viable cell counts but also actively promoted cell death, as evidenced by trypan blue assays and Annexin V/PI staining. Galangin appeared more cytostatic, with limited effects on cell viability. Taken together, these in vitro results reinforce the potential of ACA as a more potent anti-lymphoma agent, with galangin offering a safer alternative due to its lower toxicity profile.

Network pharmacology analysis revealed substantial overlap between galangin- or ACA-associated targets and lymphoma-related genes. PPI network analysis identifiedAKT1, ALB, HSP90AA1, EGFR, and CASP3 as central hub genes. GO and KEGG enrichment indicated involvement in phosphorylation, signal transduction, and apoptosis via the PI3K-Akt and MAPK signaling pathways, apoptosis-related signaling, and anti-inflammatory processes. These data suggest that both galangin and ACA exert multi-target anti-lymphoma effects, notably through Akt inhibition, with overlapping but distinct molecular signatures.

This study demonstrated that galangin and ACA induced dose-dependent apoptosis in Raji and Daudi cells, as evidenced by increased cl-Casp3 expression and elevated Annexin V-positive populations. Cl-Casp3, a hallmark of intrinsic apoptotic pathway, was used as a molecular marker to evaluate the pro-apoptotic effects of both compounds in lymphoma cell lines. Both compounds induced a dose-dependent increase in cl-Casp3 expression, with ACA exhibiting markedly stronger effects—particularly in Raji cells, where cl-Casp3 levels rose by 9.86-fold compared with the vehicle control at the IC_50_ concentration. This molecular response was corroborated by functional apoptosis assays using Annexin V-FITC/PI staining, which revealed a significant accumulation of both early and late apoptotic populations, especially in ACA-treated cells. Notably, ACA induced apoptosis in 46.03 ± 4.10% of Raji cells and 24.53 ± 6.86% of Daudi cells at their respective IC_50_ concentrations, whereas galangin exhibited a more modest pro-apoptotic effect, primarily in Daudi cells. This differential sensitivity may be attributed to the inherent differences in Bcl-2 expressions between the two cell lines [40]. Raji cells, which typically express higher levels of the anti-apoptotic protein Bcl-2, may require stronger pro-apoptotic stimuli to undergo cell death. In contrast, Daudi cells, with lower Bcl-2 expression, may be more susceptible to apoptosis, thereby explaining their distinct responses to galangin and ACA. Furthermore, the strong correlation between cl-Casp3 expression and the percentage of apoptotic cells confirmed the involvement of the intrinsic apoptotic pathway. In addition, crude galangal extract at its IC_20_ concentration exhibited the activities consistent with those of tis active compound. It suppressed c-Myc protein expression, inhibited cell cycle progression, and induced apoptosis in Raji and Daudi cells, as shown in Appendix A. These results are supported by prior studies on galangin’s pro-apoptotic mechanisms. Galangin has been reported to inhibit cell cycle progression and enhance apoptosis via PTEN activation and Casp3-mediated pathways in retinoblastoma cells [41]. In triple-negative breast cancer, galangin enhanced TRAIL-induced apoptosis by activating Casp3 via the AMPK signaling pathway [42]. Such findings reinforce galangin’s role in modulating Casp3 across cancer type. ACA has likewise demonstrated broad pro-apoptotic activity. Studies in myeloid leukemia (NB4) cells showed that ACA induced apoptosis via ROS generation and Casp3 activation [36]. Additionally, ACA has been reported to trigger apoptosis in glioblastoma cells through enhanced Casp3 activity, as well as in Ehrlich ascites tumor models [43]. Taken together, the mechanistic schematic diagram in Figure 6 illustrates the both galangin and ACA modulate c-Myc through the PI3K, Akt, and MAPK signaling pathways, thereby suppressing cell proliferation and promoting apoptosis.

Overall, these findings highlight ACA as a potent apoptosis-inducing agent in lymphoma cells, acting primarily through mitochondrial pathways and c-Myc suppression, whereas galangin offers a safer alternative with cytostatic and immune-modulating properties. These mechanistic insights underscore the therapeutic potential of Zingiberaceae-derived compounds as complementary agents in lymphoma treatment. In line with this, previous studies have also demonstrated that ACA possesses broad pharmacological activities, including anticancer effects mediated through AMPK and HER2 signaling pathways. Specifically, ACA was shown to inhibit proliferation and invasion while promoting apoptosis in endocrine-resistant breast cancer cells, with in vivo efficacy confirmed in both zebrafish and mouse models [44,45]. Consistently, ethanolic extracts of *A. galanga* containing galangin and ACA significantly reduced tumor volume and Ki-67 expression in C3H mice bearing breast adenocarcinoma in a dose-dependent manner, supporting ACA as the principal bioactive compound responsible for these anticancer effects [46].

## 5. Conclusions

This study highlights the promising anti-lymphoma potential of galangin and ACA, two bioactive compounds derived from *Alpinia galanga*, a member of the Zingiberaceae family. Both compounds exhibited dose-dependent cytotoxicity in Raji and Daudi lymphoma cell lines, with ACA demonstrating markedly greater potency but a narrower selectivity index compared to galangin. Molecular analyses revealed that neither compound induced significant cell cycle arrest at IC_20_ concentrations, suggesting alternative mechanisms of action. Notably, both galangin and ACA effectively downregulated c-Myc and p-c-Myc expression, modulated apoptosis-related pathways, and induced Casp3–mediated cell death. Network pharmacology and KEGG pathway analyses further supported their involvement in critical oncogenic pathways, particularly the PI3K-Akt and MAPK signaling cascades. Collectively, these findings provide mechanistic insight into their anticancer activities and lay the groundwork for future preclinical development of galangin and ACA as multi-targeted therapeutic agents in lymphoma. Emphasis should be placed on dosage optimization and advanced delivery strategies to enhance therapeutic efficacy while minimizing off-target toxicity. It should be noted that this study represents only an in vitro finding, and further in vivo validation is warranted before translating these results into potential future therapies.

## Figures and Tables

**Figure 1 biology-14-01098-f001:**
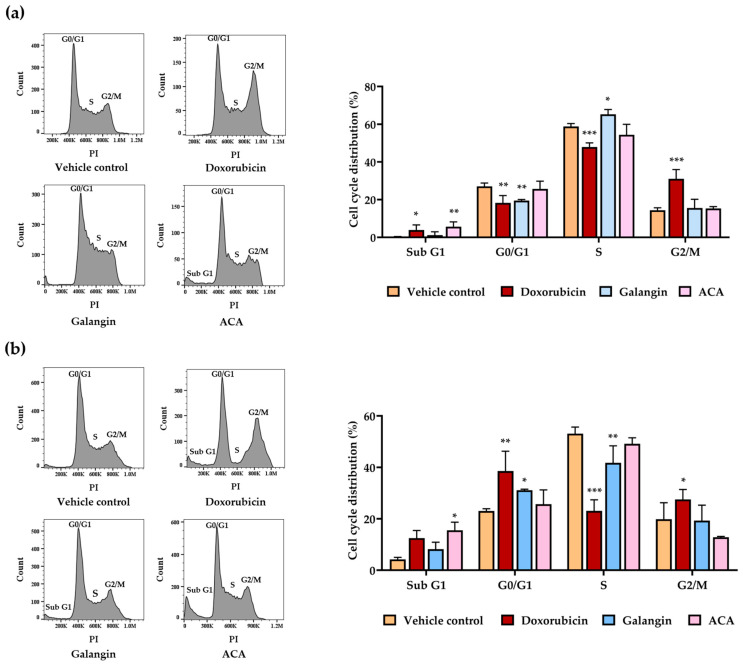
Effect of galangin and ACA Cell Cycle in Raji and Daudi Cells (**a**) Raji cells at density of 1 × 10^5^ cells/mL treated with galangin and ACA at IC_20_ concentrations for 48 h, doxorubicin was used as positive control. (**b**) Daudi cells at density of 2 × 10^5^ cells/mL treated with galangin and ACA at IC_20_ concentration for 48 h, doxorubicin was used as positive control. Data are expressed as mean ± SD of three independent experiments. Asterisks (*) denote significant differences from vehicle control (* *p* < 0.05, ** *p* < 0.01, *** *p* < 0.001).

**Figure 3 biology-14-01098-f003:**
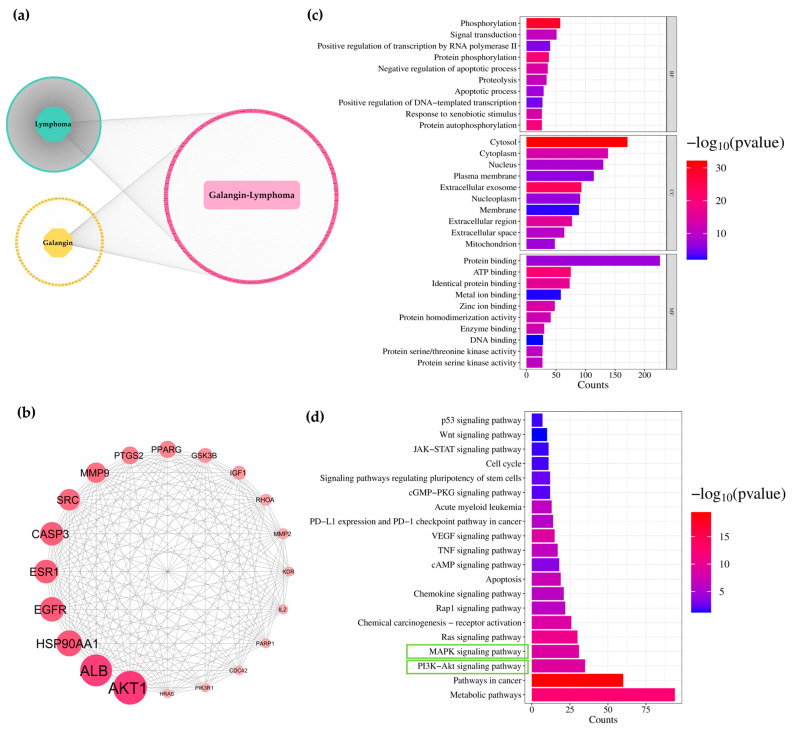
Predicted targets of galangin against lymphoma using network pharmacology. (**a**) A lymphoma-galangin targets network. The green ellipse represents gene targets related only to lymphoma. Yellow ellipse stands for gene targets related only to galangin. Pink circular layout represent gene targets related to both galangin and lymphoma. (**b**) Protein–protein interaction (PPI) network of the top 20 key gene targets of galangin against lymphoma. (**c**) Gene ontology (GO) enrichment analysis of the overlapping gene targets. The top 10 terms in biological processes (BP), cellular components (CC), and molecular functions (MF) are shown. (**d**) KEGG pathway enrichment analysis showing the top 20 signaling pathways associated with the potential gene targets of galangin against lymphoma. The green box highlights the signaling pathway with the highest gene count.

**Figure 4 biology-14-01098-f004:**
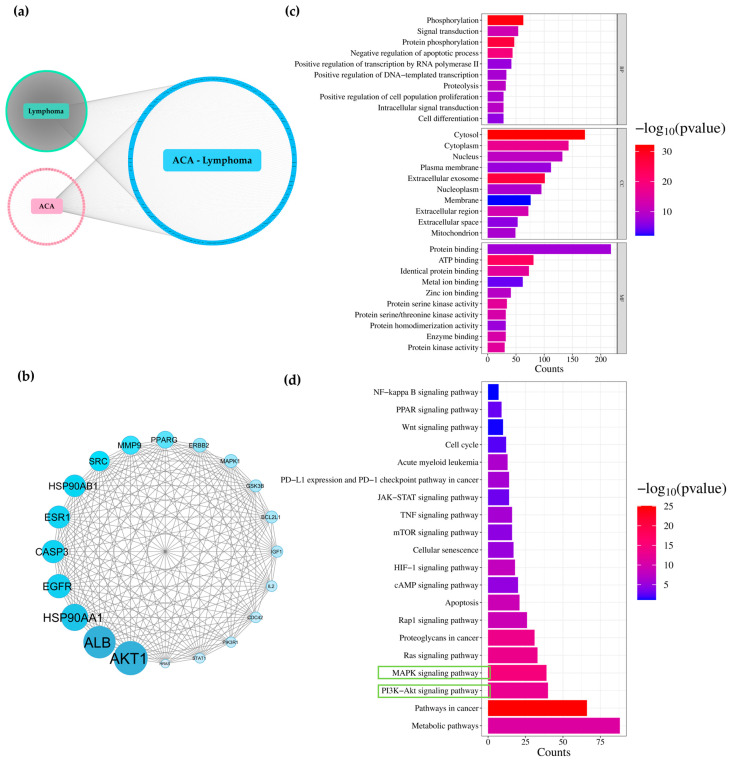
Predicted targets of ACA against lymphoma using network pharmacology. (**a**) A lymphoma-ACA targets network. The green ellipse represents gene targets related only to lymphoma. Pink ellipse stands for gene targets related only to ACA. Blue circular layout represents gene targets related to both ACA and lymphoma. (**b**) Protein–protein interaction (PPI) network of the top 20 key gene targets of ACA against lymphoma. (**c**) Gene ontology (GO) enrichment analysis of the overlapping gene targets. The top 10 terms in biological processes (BP), cellular components (CC), and molecular functions (MF) are shown. (**d**) KEGG pathway enrichment analysis showing the top 20 signaling pathways associated with the potential gene targets of ACA against lymphoma. The green box highlights the signaling pathway with the highest gene count.

**Figure 6 biology-14-01098-f006:**
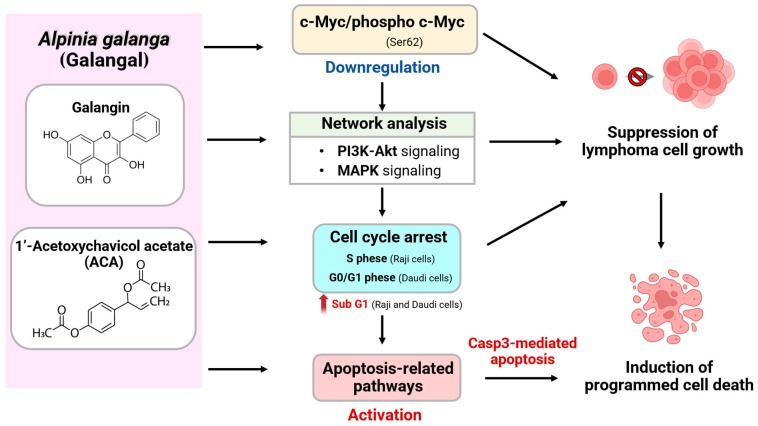
Proposed mechanisms underlying the anti-lymphoma effects of galangin and ACA from galanga. Galangin and ACA exert anti-proliferative and pro-apoptotic effects on Raji and Daudi lymphoma cells through modulation of multiple signaling pathways. Both compounds downregulate c-Myc and p-c-Myc (Ser62), as identified by network analysis implicating PI3K-Akt and MAPK signaling as upstream regulators. These molecular alterations lead to cell cycle arrest—at the S phase in Raji cells, the G0/G1 phase in Daudi cells, and an increase in the sub G1 population in both cell lines–followed by activation of apoptosis-related pathways, including aspace-3 (Casp3)-mediated apoptosis. Collectively, these events contribute to suppression of lymphoma cell growth and induction of programmed cell death.

**Table 1 biology-14-01098-t001:** Inhibitory concentration values at 50% growth (IC_50_ values) and selectivity index (SI) of crude ethanolic extracts from Zingiberaceae plants on Raji and Daudi, and PBMCs after incubation for 48 h.

Zingiberaceae Plant	IC_50_ Value (μg/mL) (Mean ± SD)			Selectivity Index (SI)	
	Raji	Daudi	PBMCs	Raji	Daudi
Galangal (*Alpinia galanga*)	31.52 ± 3.59	14.20 ± 2.34	43.18 ± 3.05	1.36	3.04
Black turmeric (*Curcuma aeroginosa*)	55.54 ± 5.74	35.48 ± 3.23	79.11 ± 3.92 ^#^	1.42	2.23
Black ginger (*Kaempferia parviflora*)	46.30 ± 3.97	17.47 ± 0.73	46.77 ± 1.48 ^#^	1.01	2.68
Phlai lueang (*Zingiber montanum*)	43.67 ± 5.25	22.00 ± 0.98	56.25 ± 3.41 ^#^	1.28	2.56
Phlai dam (*Zingiber ottensii*)	51.74 ± 6.77	32.65 ± 1.42	>100 ^#^	>1.93	>3.06

^#^ Values from our previous report [26], obtained using the same batches of crude ethanolic extracts from Zingiberaceae plants, were used to calculate the selectivity index.

**Table 2 biology-14-01098-t002:** Inhibitory concentration values at 50% growth (IC_50_ values) and selectivity index (SI) of active compounds in Raji, Daudi, and PBMCs after incubation for 48 h.

Compound	IC_50_ Value			Selectivity Index (SI)	
	Raji	Daudi	PBMCs	Raji	Daudi
Galangin (μg/mL)	21.00 ± 1.58	10.75 ± 1.29	>100	>4.76	>9.30
ACA (μg/mL)	1.93 ± 0.26	1.74 ± 0.46	4.69 ± 0.25	2.43	2.70
Doxorubicin (ng/mL)	61.00 ± 12.86	20.79 ± 2.20	>1000	>16.39	>48.10

## Data Availability

The original contributions presented in the study are included in the article/Appendix A, further inquiries can be directed to the corresponding authors.

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
