# Peer review of "Galangin and 1′-Acetoxychavicol Acetate from Galangal (Alpinia galanga) Suppress Lymphoma Growth via c-Myc Downregulation and Apoptosis Induction"

_biology, 2025, doi:10.3390/biology14081098_

Round 1

Reviewer 1 Report

Comments and Suggestions for Authors

The manuscript ‘Galangin and 1’-Acetoxychavicol Acetate from Galangal (Alpinia galanga) Suppress Lymphoma Growth via c-Myc Downregulation and Apoptosis Induction’ is a well-structured paper aimed at identifying the dose-dependent anti-cancer effect of galangin and ACA on Burkitt’s Lymphoma cells. The introduction is very well written, addressing the rationale and having a good flow for the readers. The quality of the paper can be further improved by addressing/incorporating the following minor points:

  1. In the Abstract, please correct ‘ant-lymphoma’ to ‘anti-lymphoma’ in Line 42.
  2. Add a mechanistic schematic diagram depicting the upstream/downstream targets for galangin and ACA to achieve suppress growth and induce apoptosis in cancer cells. Including this in the discussion section would increase the impact of these findings and provide a bigger picture of the mechanism involved.
  3. Line 179-180: Please mention at cell density at the time of seeding in 96-well plates.
  4. Fig 3c & 3d and Fig.4c & 4d: encircle or highlight the names of the pathways that are upregulated/downregulated. Also, the numerical values on the Log scale are not visible Please improve the font.
  5. Emphasize in the conclusion section that this study is merely an in vitro finding and also throw some light on the need for further in vivo validation before translating them into future therapies.
Comments on the Quality of English Language

Minor grammatical and formatting issues are present in the manuscript. 

Author Response

Reviewer 1

Comments and Suggestions for Authors

The manuscript ‘Galangin and 1’-Acetoxychavicol Acetate from Galangal (Alpinia galanga) Suppress Lymphoma Growth via c-Myc Downregulation and Apoptosis Induction’ is a well-structured paper aimed at identifying the dose-dependent anti-cancer effect of galangin and ACA on Burkitt’s Lymphoma cells. The introduction is very well written, addressing the rationale and having a good flow for the readers. The quality of the paper can be further improved by addressing/incorporating the following minor points:

1. In the Abstract, please correct ‘ant-lymphoma’ to ‘anti-lymphoma’ in Line 42.

Response: ‘ant-lymphoma’ has been changed to ‘anti-lymphoma’ on page 1, line 42.

2. Add a mechanistic schematic diagram depicting the upstream/downstream targets for galangin and ACA to achieve suppress growth and induce apoptosis in cancer cells. Including this in the discussion section would increase the impact of these findings and provide a bigger picture of the mechanism involved.

Response: A mechanistic schematic diagram illustrating the upstream and downstream targets of galangin and ACA-specifically PI3K, Akt, MAPK, and c-Myc-leading to suppressed growth and induction of apoptosis in lymphoma cells is presented in Figure 6 (page 21). The corresponding description has been incorporated into the Discussion section to enhance the overall impact of these findings (page 20, lines 773–776), as follows:
“Taken together, the mechanistic schematic diagram in Figure 6 illustrates the both galangin and ACA modulate c-Myc through the PI3K, Akt, and MAPK signaling pathways, thereby suppressing cell proliferation and promoting apoptosis.”

3. Line 179-180: Please mention at cell density at the time of seeding in 96-well plates.

Response: Cell density at the time of seeding in 96-well plates (1.0 ´ 105 cells/ml for Raji cells and 2.0 ´ 105 cells/ml for Daudi cells) have been added on page 4, line 183.

4. Fig 3c & 3d and Fig.4c & 4d: encircle or highlight the names of the pathways that are upregulated/downregulated. Also, the numerical values on the Log scale are not visible Please improve the font.

Response: Figures 3c and 4c (pages 14-15): In the GO enrichment analysis, the most enriched terms in the Biological Process, Cellular Component, and Molecular Function categories were “phosphorylation,” “cytosol,” and “protein binding,” respectively. The high gene counts in these categories indicate that galangin- and ACA-related lymphoma targets are predominantly cytosolic proteins involved in phosphorylation-dependent signal transduction and extensive protein–protein interactions, suggesting a broad regulatory impact on intracellular signaling networks. In the KEGG pathway enrichment analysis (Figures 3d and 4d), the PI3K–Akt and MAPK signaling pathways exhibited the highest gene counts, indicating that a substantial proportion of galangin- and ACA-related lymphoma targets are involved in these pathways. This suggests that modulation of PI3K–Akt and MAPK signaling (highlighted with green boxes in the Figures 3d and 4d) may represent the primary mechanisms underlying galangin’s and ACA’s anti-lymphoma activity. However, the network analysis used here does not provide information on whether the expression of specific proteins or pathways is upregulated or downregulated; therefore, it is not possible to encircle or highlight such information in these figures. The numerical values on the log scale in Figures 3d and 4d (pages 14 and 15) have been enlarged to improve visibility.

5. Emphasize in the conclusion section that this study is merely an in vitrofinding and also throw some light on the need for further in vivo validation before translating them into future therapies.

Response: “It should be noted that this study represents only an in vitro finding, and further in vivo validation is warranted before translating these results into potential future therapies.” has been added in the conclusion on page 21, lines 797-799.

Comments on the Quality of English Language

Minor grammatical and formatting issues are present in the manuscript. 

Response: We have carefully re-checked and corrected all minor grammatical and formatting issues, with all modifications shown using track changes.

Reviewer 2 Report

Comments and Suggestions for Authors

Dear Authors

In the article “Galangin and 1’-Acetoxychavicol Acetate from Galangal (Alpinia galanga) Suppress Lymphoma Growth via c-Myc Down-regulation and Apoptosis Induction”, by Sirinya Moakmamern et al., the authors have explored the therapeutic potential of traditional Zingiberaceae plants - galangal (Alpinia galanga), black turmeric (Curcuma aeroginosa), black ginger (Kaempferia parviflora), phlai lueang (Zingiber montanum), and phlai dum (Zingiber ottensii), against lymphoma using lymphoma cell lines (Raji and Daudi). The authors show that galangal exhibits the best anti-cancer potential amongst the above-mentioned plants, and further have focused on its active compounds – Galangin and ACA. The authors demonstrate that the two compounds exhibit strong anti-cancer potential, c-Myc downregulation, and induction of apoptosis. The results are consistent with the existing reports on the mode of action of galangal compounds on other cancer types and present the findings in context with lymphoma. This study highlights the promising anti-lymphoma potential of galangin and ACA, from Alpinia galanga (Galangal), which exhibit a dose-dependent cytotoxicity, in vitro.

Certain observations have been made for the manuscript that need to be addressed:

  1. The Materials and Methods section could be made crisper and more unnecessary theoretical portion can be removed/ limited, e.g., detailed principles. details of the blood components, etc.
  2. Section 2.5.2. – The chronology of the sentences is not proper.
  3. The authors have claimed that the effects of Zingiberaceae plant extracts and their active components on cell cycle arrest, western blot, annexin V staining, etc. However, only the results for the active components have been listed. The results are not added in the manuscript or even in the supplementary. The same must be done.
  4. Section 3.1 – The Authors claim that “In the next experiment, the active compounds will be identified and further investigated for their biological activities”. However, no such identification procedures have been defined, and the active compounds have been chosen based on existing literature. The line needs to be modified appropriately.
  5. The selectivity index for the crude extracts must be calculated, as this can indicate the potential of other extracts that have been left out solely on the basis of IC50
  6. The authors claim that a selectivity index >1 is good enough for the selection of compounds. However, traditionally, the selectivity index >3 is generally considered to signify a good level of selectivity and is desirable for a drug candidate. The SI of ACA is 2.43 and 2.70 in Raji and Daudi cell lines, respectively, which may not be desirable.
  7. Section 3.3, 367 – Understanding to be replaced with Underlying.
  8. Section 3.4 – The authors have claimed that “extending the treatment to 48 h in Daudi cells resulted in excessive cell death and an insufficient number of viable cells for analysis”. However, the Daudi cells were treated for 48 hours in other experiments, and the results/ counts do not seem to be affected (Fig. 1 (a) and (b)). Also, the seeding densities for the two cell lines seem to be different. The authors need to repeat the experiments.
  9. Fig. 2 (a) and (b) – The normalization controls (GAPDH) used in the experiment do not seem to be consistent. It is felt that there is unequal loading in the wells. The experiments need to be re-examined.
Comments on the Quality of English Language
  1. The language and grammar need urgent attention.
  2. Apart from the minor grammatical changes in the manuscript, the manuscript ought to be written in the singular tense to make the manuscript more lucid and interesting.
  3. The chronology of the lines in the materials and methods section needs to be re-examined.

Author Response

Reviewer 2

Comments and Suggestions for Authors

In the article “Galangin and 1’-Acetoxychavicol Acetate from Galangal (Alpinia galanga) Suppress Lymphoma Growth via c-Myc Down-regulation and Apoptosis Induction”, by Sirinya Moakmamern et al., the authors have explored the therapeutic potential of traditional Zingiberaceae plants - galangal (Alpinia galanga), black turmeric (Curcuma aeroginosa), black ginger (Kaempferia parviflora), phlai lueang (Zingiber montanum), and phlai dum (Zingiber ottensii), against lymphoma using lymphoma cell lines (Raji and Daudi). The authors show that galangal exhibits the best anti-cancer potential amongst the above-mentioned plants, and further have focused on its active compounds – Galangin and ACA. The authors demonstrate that the two compounds exhibit strong anti-cancer potential, c-Myc downregulation, and induction of apoptosis. The results are consistent with the existing reports on the mode of action of galangal compounds on other cancer types and present the findings in context with lymphoma. This study highlights the promising anti-lymphoma potential of galangin and ACA, from Alpinia galanga (Galangal), which exhibit a dose-dependent cytotoxicity, in vitro.

Certain observations have been made for the manuscript that need to be addressed:

1. The Materials and Methods section could be made crisper and more unnecessary theoretical portion can be removed/ limited, e.g., detailed principles. details of the blood components, etc.

Response: The details of principles of blood components have already been removed from the Materials and Methods section already (page 5, lines 210-218).

2. Section 2.5.2. – The chronology of the sentences is not proper.

Response: Section 2.5.2 has been revised to ensure proper chronology of the sentences, as shown on page 6, lines 236-238.

3. The authors have claimed that the effects of Zingiberaceaeplant extracts and their active components on cell cycle arrest, western blot, annexin V staining, etc. However, only the results for the active components have been listed. The results are not added in the manuscript or even in the supplementary. The same must be done.

Response: In the initial screening, several Zingiberaceae plant extracts were evaluated for cytotoxicity, and galangal (Alpinia galanga) exhibited the most potent activity. Based on these findings, the present study focused on further investigating the biological activities of its major active constituents, galangin and ACA. Nevertheless, to address the reviewer’s comment, the results for the galangal crude ethanolic extract have now been included in the Supplementary Data (Figure S3) and described in the Discussion section (page 20, lines 764–767) for comparison with its active compounds as follows:

“In addition, galangal crude ethanolic extract at its IC20 concentration exhibited the activities consistent with those of tis active compound. It suppressed c-Myc protein expression, inhibited cell cycle progression, and induced apoptosis in Raji and Daudi cells, as shown in Figure S3.”

4. Section 3.1 – The Authors claim that “In the next experiment, the active compounds will be identified and further investigated for their biological activities”. However, no such identification procedures have been defined, and the active compounds have been chosen based on existing literature. The line needs to be modified appropriately.

Response: Thank you very much for your kind suggestions. We have modified the sentence clarifying the source of the major active compounds, galangin and ACA, on page 8, lines 345-348, as follows:

“Based on previous literature [27-29], galangin and acetoxychavicol acetate (ACA) were selected as the major active constituents of galangal and were further investigated for their biological activities in this study.”

5. The selectivity index for the crude extracts must be calculated, as this can indicate the potential of other extracts that have been left out solely on the basis of IC50

Response: The selectivity index (SI) for the crude extracts has been calculated and is now included in Table 1. In this study, the crude extract from galangal (Alpinia galanga) was tested in PBMCs. The ICâ‚…â‚€ values for black turmeric (Curcuma aeruginosa), black ginger (Kaempferia parviflora), Phlai lueang (Zingiber montanum), and Phlai dam (Zingiber ottensii) were obtained from our previous study (Panyajai, et al., 2025), in which the same batches of crude ethanolic extracts were used. However, this information has also been specified in the footnote of Table 1 for clarification (page 8, lines 349-353).

References

Panyajai P, Viriyaadhammaa N, Chiampanichayakul S, Sakamoto Y, Okonogi S, Moroishi T, Anuchapreeda S. Anticancer and cancer preventive activities of shogaol and curcumin from Zingiberaceae family plants in KG-1a leukemic stem cells. BMC Complement Med Ther. 2025;25(87). https://doi.org/10.1186/s12906-025-04829-7.

6. The authors claim that a selectivity index >1 is good enough for the selection of compounds. However, traditionally, the selectivity index >3 is generally considered to signify a good level of selectivity and is desirable for a drug candidate. The SI of ACA is 2.43 and 2.70 in Raji and Daudi cell lines, respectively, which may not be desirable.

Response: Thank you for your valuable comment. While a selectivity index (SI) >3 is traditionally considered to indicate a good level of selectivity for a drug candidate, several studies have reported the use of SI >1 as a threshold for desirable selectivity against cancer cells. In this study, we adopted the latter criterion, which is defined as the ratio of the ICâ‚…â‚€ value in healthy cells to that in cancer cells. The following references illustrate the use of this criterion:

References

1) Panyajai P, Viriyaadhammaa N, Chiampanichayakul S, Sakamoto Y, Okonogi S, Moroishi T, Anuchapreeda S. Anticancer and cancer preventive activities of shogaol and curcumin from Zingiberaceae family plants in KG-1a leukemic stem cells. BMC Complement Med Ther. 2025;25(87). https://doi.org/10.1186/s12906-025-04829-

2) Indrayanto, G.; Putra, G.S.; Suhud, F. Chapter six - Validation of in-vitro bioassay methods: application in herbal drug research. In Profiles of Drug Substances, Excipients and Related Methodology, Al-Majed, A.A., Ed. Academic Press: USA, 2021; Vol. 46, pp 273-307.

3) Lica, J.J.; Wieczór, M.; Grabe, G.J.; Heldt, M.; Jancz, M.; Misiak, M.; Gucwa, K.; Brankiewicz, W.; Maciejewska, N.; Stupak, A., et al. Effective drug concentration and selectivity depends on fraction of primitive cells. Int J Mol Sci. 2021, 22, 4931, doi:10.3390/ijms22094931.

4) Kaminsky, R.; Schmid, C.; Brun, R. An “in vitro selectivity index” for evaluation of cytotoxicity of antitrypanosomal compounds. In Vitro Toxicol. 1996, 9, 315-324.

5) Badisa, R.B.; Darling-Reed, S.F.; Joseph, P.; Cooperwood, J.S.; Latinwo, L.M.; Goodman, C.B. Selective cytotoxic activities of two novel synthetic drugs on human breast carcinoma MCF-7 cells. Anticancer Res. 2009, 29, 2993-2996.

6) De Oliveira, P.F.; Alves, J.M.; Damasceno, J.L.; Oliveira, R.A.; Dias, H., Jr.; Crotti, A.E.; Tavares, D.C. Cytotoxicity screening of essential oils in cancer cell lines. Bras. Farmacogn. 2015, 25, 183-188.

7) Peña-Morán, O.A.; Villarreal, M.L.; Álvarez-Berber, L.; Meneses-Acosta, A.; Rodríguez-López, V. Cytotoxicity, Post-Treatment Recovery, and Selectivity Analysis of Naturally Occurring Podophyllotoxins from Bursera fagaroides var. fagaroides on Breast Cancer Cell Lines. 2016, 21, 1013.

7. Section 3.3, 367 – Understanding to be replaced with Underlying.

Response: Thank you very much for reviewing the writing. The term “understanding” has been replaced with “underlying” on page 9, line 382.

8. Section 3.4 – The authors have claimed that “extending the treatment to 48 h in Daudi cells resulted in excessive cell death and an insufficient number of viable cells for analysis”. However, the Daudi cells were treated for 48 hours in other experiments, and the results/ counts do not seem to be affected (Fig. 1 (a) and (b)). Also, the seeding densities for the two cell lines seem to be different. The authors need to repeat the experiments.

Response: From the analysis of the total cell number after 48 hours of treatment, Daudi cells exhibited a marked decrease in cell count due to inhibited cell proliferation, cell cycle arrest at the G0/G1 phase, and induction of cell death. The reduced cell number (Figure S2) resulted in an insufficient yield of protein after extraction, making Western blot analysis unfeasible. In contrast, other experiments, including those presented in Fig. 1(a) and (b), could still be performed because they required fewer cells or lower for detection by Flow cytometry. In Section 3.4, we used twice the number of Daudi cells compared with Raji cells, as Daudi cells have a much slower growth rate than Raji cells (Figure S1). The cell number calculations were based on the growth curve of each cell line. A 24-hour treatment period was found to be more suitable for Daudi cells, as it did not markedly induce cell death and allowed sufficient viable cells for the determination of target gene expression.

9. 2(a) and (b) – The normalization controls (GAPDH) used in the experiment do not seem to be consistent. It is felt that there is unequal loading in the wells. The experiments need to be re-examined.

Response: GAPDH was re-examined and confirmed to be consistent across all samples. The protein concentrations of all lysates were re-measured using the BCA Protein Assay Kit, and equal amounts of protein were loaded in each well to ensure consistent loading. The revised Western blot images now reflect equal loading in all lanes.

Comments on the Quality of English Language

10. The language and grammar need urgent attention.

Response: The English language and grammar have been thoroughly re-checked and revised, with all modifications shown using track changes.

11. Apart from the minor grammatical changes in the manuscript, the manuscript ought to be written in the singular tense to make the manuscript more lucid and interesting.

Response: We re-checked writing in the singular tense already.

12. The chronology of the lines in the materials and methods section needs to be re-examined.

Response: We have re-checked the manuscript to ensure consistent use of the singular tense, in addition to correcting minor grammatical issues.

Round 2

Reviewer 2 Report

Comments and Suggestions for Authors

No in vivo experiments have been reported. Apart from that the current version of the manuscripts seems to have addressed the observations made in the previous version. The results seem consistent with other reports in different cancer types, and a similar mechanism of action has been reported. 

Comments on the Quality of English Language

The language and grammar have been improved from the previous version, although I feel the manuscript can be made more lucid and interesting with expert advise.

Author Response

Response: We thank the editor for recognizing that the revised manuscript has addressed the concerns raised in the previous version. While No in vivo experiments specific to lymphoma were included, relevant in vivo studies on galangin and ACA in other cancer models have been reported. We have incorporated these findings into our revised manuscript already on pages 20-21, lines 785-793 as follows;

“In line with this, previous studies have also demonstrated that ACA possesses broad pharmacological activities, including anticancer effects mediated through AMPK and HER2 signaling pathways. Specifically, ACA was shown to inhibit proliferation and invasion while promoting apoptosis in endocrine-resistant breast cancer cells, with in vivo efficacy confirmed in both zebrafish and mouse models [45,46]. Consistently, ethanolic extracts of A. galanga containing galangin and ACA significantly reduced tumor volume and Ki-67 expression in C3H mice bearing breast adenocarcinoma in a dose-dependent manner, supporting ACA as the principal bioactive compound responsible for these anticancer effects [47].”